

# Munich permanent urban greenhouse gas column observing network

Florian Dietrich[1], Jia Chen[1], Benno Voggenreiter[1], Patrick Aigner[1], Nico Nachtigall[1], and Björn Reger[1]

[1]Environmental Sensing and Modeling, Technical University of Munich (TUM), Munich, Germany

**Correspondence:** Florian Dietrich (flo.dietrich@tum.de) and Jia Chen (jia.chen@tum.de)

**Abstract.**

In order to mitigate climate change, it is crucial to understand the urban greenhouse gas (GHG) emissions precisely as more than two third of the anthropogenic GHG emissions worldwide originate from cities. Nowadays, urban emission estimates are mainly based on bottom-up calculation approaches with high uncertainties. A reliable and long-term top-down measurement approach could reduce the uncertainty of these emission inventories significantly.

We present the world's first urban sensor network that is permanently measuring GHGs based on the principle of differential column measurements (DCM) starting in summer 2019. These column measurements are relatively insensitive to vertical redistribution of tracer masses and surface fluxes upwind of the city. Therefore, they are well-suited to quantify GHG emissions.

However, setting up such a stationary sensor network requires an automated measurement principle. We developed our own fully automated enclosure systems for measuring $CO_2$, $CH_4$ and $CO$ column-averaged concentrations with a solar-tracking Fourier Transform spectrometer (EM27/SUN) in a fully automated and long-term manner. This includes also a software that starts and stops the measurements autonomously and can be used independently from the enclosure system.

Furthermore, we demonstrate the novel applications of such a sensor network by presenting the measurement results of our five sensor systems that are deployed in and around Munich. These results include the seasonal cycle of $CO_2$ since 2015 as well as concentration gradient measurements upwind and downwind of the city. Thanks to the automation we were also able to continue the measurements during the COVID-19 lockdown in spring 2020. By correlating the $CO_2$ column concentration gradients to the traffic amount, we demonstrate that our network is well capable to detect variations in urban emissions.

The measurements from our unique sensor network will be combined with an inverse modeling framework that we are currently developing, in order to monitor urban GHG emissions over years, identify unknown emission sources and assess how effective the current mitigation strategies are. In summary, our achievements in automating column measurements of GHGs will allow researchers all over the world to establish this novel measurement approach as a new standard for determining GHG emissions.



## 1 Introduction

Climate change is one of the defining issue of our time, which affects the entire planet. For an effective reduction of the
greenhouse gas (GHG) emissions, accurate and continuous monitoring systems for local and regional scale emissions are a
prerequisite.

Especially for urban areas, which contribute to more than 70 % of GHG emissions (Gurney et al., 2015) and are therefore
hotspots, there is a lack of accurate assessments of emissions. The city emission inventories often underestimate the emissions
as there exist unknown emission sources that are not yet included in the inventories (Chen et al., 2020; Plant et al., 2019;
McKain et al., 2015).

In the recent years, several city networks were established to improve the emission monitoring. These include networks using
in-situ high precision instruments (McKain et al., 2015; Bréon et al., 2015; Xueref-Remy et al., 2018; Lamb et al., 2016) and
low-cost sensor networks deploying NDIR sensors (Kim et al., 2018; Shusterman et al., 2016). In addition, eddy covariance
flux tower measurements are used for directly inferring city fluxes (Feigenwinter et al., 2012; Helfter et al., 2011). However,
all these approaches involve some challenges for measuring urban emission fluxes, such as high sensitivity to the boundary
layer height dynamics, large variations due to mesoscale transport phenomena or the fact that they can only capture the fluxes
of a rather small area.

Column measurements have proven to be a powerful tool for assessing GHG emissions from cities and local sources, because
they are relatively insensitive to the dynamics of the boundary layer height and to surface fluxes upwind of the city if a
differential approach is used (Chen et al., 2016). Therefore, this method has recently been widely deployed for emission studies
of cities and local sources using mass balance or other modeling techniques. In St. Petersburg, Makarova et al. (2020) deployed
2 compact solar-tracking Fourier-transform infrared (FTIR) spectrometers (EM27/SUN) and a mass balance approach to study
the emissions from the fourth largest European city. The EM27/SUN spectrometer has been developed by KIT in collaboration
with Bruker and is commercially available since 2014 (Gisi et al., 2011, 2012; Hase et al., 2016). Hase et al. (2015) and
Zhao et al. (2019) used the measurements of 5 EM27/SUNs to measure the Berlin city emissions of $CO_2$ and $CH_4$. With a
similar sensor configuration, Vogel et al. (2019) studied the Paris metropolitan area and applied the CHIMERE-CAMS model
to show that the measured concentration enhancements are mainly due to fossil fuel emissions. Jones et al. (2018) combined
the Indianapolis city measurements (5 EM27/SUNs) with an adapted inverse modeling technique to determine the urban GHG
emissions.

Besides these urban studies, column measurements are also used to investigate local sources: Chen et al. (2016) and Viatte
et al. (2017) determined the source strength of dairy farms in Chino, California. By combining column measurements with
a computational fluid dynamics (CFD) model, Toja-Silva et al. (2017) verified the emission inventory of the largest gas fired
power plant in Munich. With mobile setups, Butz et al. (2017) studied emissions from the volcano Mt. Etna, Luther et al.
(2019) quantified the coal mine emissions from upper Silesia and Klappenbach et al. (2015) utilized column measurements
on a research vessel for satellite validations above the ocean. However, these studies are all based on the campaign mode
and not suited for monitoring the urban emissions permanently. Only TCCON (Total Carbon Column Observing Network,





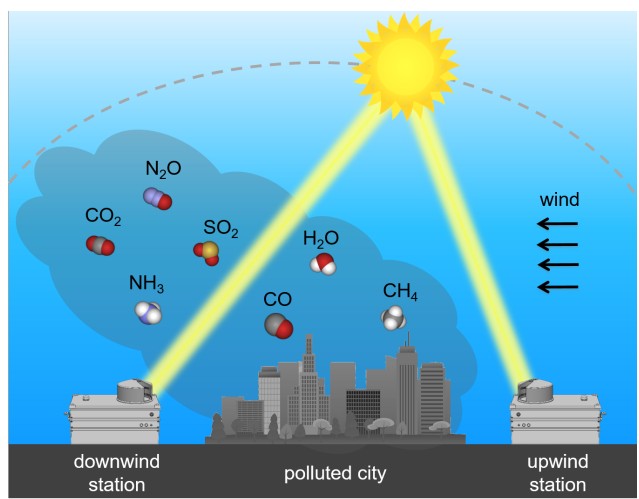

**Figure 1.** Basic principle of the differential column measurements: with the help of an upwind and at least one downwind station, the column-averaged GHG concentrations are measured. The differences between the two stations are representative for the emissions generated in the city.

Wunch et al. (2011)) and COCCON (Collaborative Carbon Column Observing Network, Frey et al. (2019); Sha et al. (2019)) are measuring the global GHG column concentrations permanently. For this purpose, TCCON uses IFS 125HR spectrometers (resolution: $0.0009\,\mathrm{cm^{-1}}$), while COCCON uses calibrated EM27/SUN spectrometers (resolution: $0.5\,\mathrm{cm^{-1}}$). Both networks
60 focus, however, on detecting GHG background concentrations and are not mainly designed to study urban emissions.

In this paper, we present the world's first permanent urban GHG network based on the differential column measurements (DCM) principle. The network is located in Munich and consists of 5 fully automated FTIR spectrometers. By combining the network with a modelling framework, we will be able to monitor urban GHG emissions over years, identify unknown emission sources, validate satellite-based GHG measurements as well as to assess how effective the current mitigation strategies are.

65 **2 Measurement principle**

As measurement principle, the DCM method is used (Chen et al., 2016). DCM is an effective approach to determine the emissions of large area sources using just a small number of stationary ground-based instruments. The basic principle of DCM is illustrated in Fig. 1. The column-averaged concentrations of a gas in the atmosphere are measured upwind and downwind of an emission source, utilizing ground-based FTIR spectrometers that use the sun as light source. The concentration enhancements
70 between the two stations are caused by the urban emissions. Chen et al. (2016) have shown that the differences between the upwind and downwind column concentrations are relatively insensitive to the boundary layer height and upstream influences. Therefore, DCM in combination with a wind-driven atmospheric transport model can be used to determine emissions.





## 3 Measurement system

In order to use the DCM principle for long-term monitoring of the urban GHG emissions, a fully automated measurement system is needed. Therefore, we developed an electronically controlled enclosure system including the related software.

### 3.1 Hardware

The enclosure system protects the spectrometer inside against harsh weather conditions and other harmful events such as power or sensor failures. Furthermore, it enables a communication between the devices inside the system and allows the host to remote control the measurements over the internet. At suited measurement conditions, such as sunny weather and valid sun elevations, the system automatically starts the measurement process. During the day, the measurements are checked regularly by the enclosure software to detect and solve malfunctions autonomously. When the measurement conditions are not suited anymore, the system stops the measurements and closes the cover to secure the spectrometer. An operator is informed about any unexpected behavior via email notification.

#### 3.1.1 Standard edition

The described enclosure is based on our first prototype system presented in Heinle and Chen (2018), which is continuously running on the rooftop of the Technical University of Munich (TUM) in the inner city of Munich since 2016. This system was developed to semi-automate the measurement process using an EM27/SUN spectrometer throughout the years. For the permanent urban GHG network, we improved this system to make it more reliable, easier to transport and fully autonomous.

Our new enclosure system is based on a lightweight yet robust aluminum housing (Zarges K470 box, waterproof according to IP54) that we modified for our purposes. The CAD model of this system is shown in Fig. 2. A rotating cover at the top of the housing allows the sunrays to hit the mirrors of the solar tracker at arbitrary azimuth and elevation angles. Every 10 degrees a magnet is fixed in the outer cover (see Fig. 3). Reed sensors in the inner cover are counting these signals so that the relative position of the cover can be computed. Before opening the cover and after every full rotation, two additional Reed sensors are indicating the absolute zero position. The target position of the cover is computed automatically depending on the coordinates of the site and the time. Optical rain and direct solar radiation sensors indicate whether the current environmental conditions are suited for measurements.

Signal lamps, push buttons and an emergency stop button can be used to control the basic functions of the enclosure directly at the site. Full control can only be achieved by remote access to the enclosure computer, which is an industrial embedded box PC. In addition to the remote access, the computer is also responsible for controlling the spectrometer and the solar tracker and to store the interferograms before they are transferred to our retrieval cloud via the internet.

The enclosure system itself is controlled by a Siemens S7-1200 PLC (programmable logic controller) and not by the enclosure computer that runs with a Microsoft Windows operating system. This approach ensures that the safety features such as rain or power failure detection, cover motor control, temperature control, etc. are separated from the Windows operating system, making the enclosure less error-prone and fail-safe.





All the additional electronics are placed in the rear part of the enclosure systems and are shown in Fig. 4 more in detail. Besides the PLC we installed an LTE router, a heater, the motor driver, two circuit breakers, surge protection devices and an RCCB (residual current circuit breaker). In addition, new relays were added to the system to be able to reset all error-prone devices such as the computer, the router or the PLC remotely. In order to make the system as lightweight as possible, we replaced the large and heavy thermo-electrical cooler (TEC) by a cooling fan and a heater, and replaced the lead-acid
battery of the UPS (uninterruptible power supply) by a capacitor-based energy storage. All the devices inside the system are communicating over the two standard protocols TCP/IP and USB.

A photo of one of the four newly developed enclosure systems for the Munich network can be seen in Fig. 5. It shows the measurement setup at our southern site on top of a flat rooftop.

### 3.1.2    Universal editions

Our enclosure system was originally developed to measure the GHG concentrations in Munich at a latitude of 48.15° N. Therefore, the rotating cover that protects the solar tracker from bad weather was designed to enable measurements for all possible solar angles at such a latitude. If the enclosure system is, however, used somewhere else in the world, these limitations need to be considered. That is why we designed our new cover so that it can measure solar elevation angles up to about 80° and azimuth angles between 30° and 300°, which covers most of the places worldwide. Furthermore, we adapted some features
to overcome challenges such as extreme temperatures as well as high relative humidity. We developed two of these special editions and tested them both at very low and high latitudes; one in Uganda next to the equator and one in Finland next to the polar circle.

As part of the NERC MOYA project, the University of Leicester is using our enclosure system to measure $CH_4$ emissions from the wetlands north of Jinja, Uganda (latitude: 0.4° N) since the beginning of 2020 (Boesch et al., 2018; Humpage et al.,
2019). Besides significant higher temperatures and relative humidity in comparison to Munich, the very high solar elevation angles (up to 90°) are challenging. These high angles are a problem for both the cover of the enclosure as it is blocking the sun in such cases and for the solar tracker of the spectrometer. The solar tracker of the EM27/SUN can only measure up to elevation angles of about 85°. For higher elevations, the control algorithm is not stable anymore. Therefore, both the spectrometer and the cover cannot work properly at such high elevation angles.

To overcome this challenge, we tilted the whole enclosure system by a few degrees to simulate that the instrument is located at a site with a higher latitude than it actually is. This is done using two state-of-the-art car jacks (see Fig. 6), which can elevate the side of the enclosure that points towards the equator up to 15°. In this way, the very low elevation angles cannot be measured anymore, as the sun is then blocked by the lid of the enclosure, which is however not an issue. This is because the air mass dependency of the slant column cannot be reliably handled by the GFIT retrieval algorithm at these high solar zenith
angles (Wunch et al. 2011). Using this unique approach, both the solar tracker and the rotating cover work properly at high elevation angles, which makes this approach suited for locations at low latitudes.





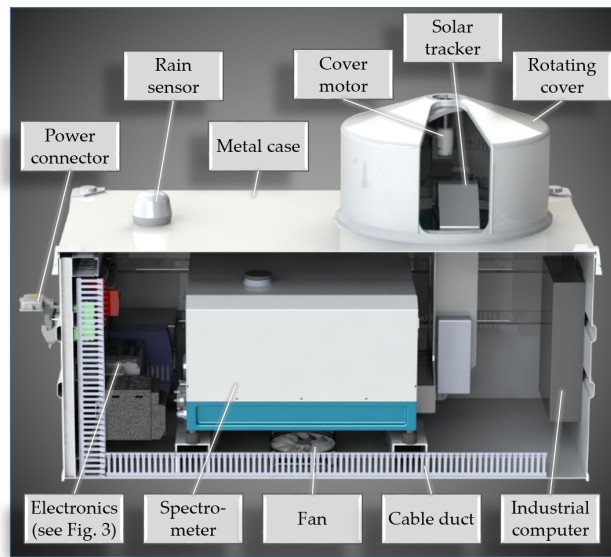

**Figure 2.** Side view of the enclosure system (CAD model)

Since the temperature and relative humidity are much higher than in central Europe, the normally used fan and heater are replaced by two 150 W theromoelectrical coolers. They cannot only control the temperature to a constant level of 25 °C under normal weather conditions in Uganda but also condensate water vapour to reduce the relative humidity inside the system.

Our enclosure is, however, not only suited to work at very low latitudes but also at high ones. To test the system under such conditions, we built another enclosure system for the COCCON site next to the TCCON station in Sodankylä at a latitude of 67.4° N (Tu et al. 2020). There the system is continuously measuring since 2018, which shows that our system cannot only withstand cold winters but is also suited to measure a large azimuth angle range.

Overall, we developed a system that is universally applicable and can be used for a wide latitude range to enable ground-
based GHG measurements worldwide with minimum effort and maximum measurement data.

## 3.2 Software

For controlling and automating the enclosure system, two independent software were developed: one is to control all safety and enclosure features that are monitored by the PLC (ECon) and the other one is to control the spectrometer and automatically perform the measurements (Pyra). Latter one also includes a user interface (UI) where the operator can set all parameters and
observe the current state of the system.

### 3.2.1 Enclosure control (ECon)

The enclosure control software ECon was already a part of the first enclosure version (Heinle and Chen, 2018). There, a microcontroller program is used to control the enclosure features such as opening and closing the rotating cover, analyzing


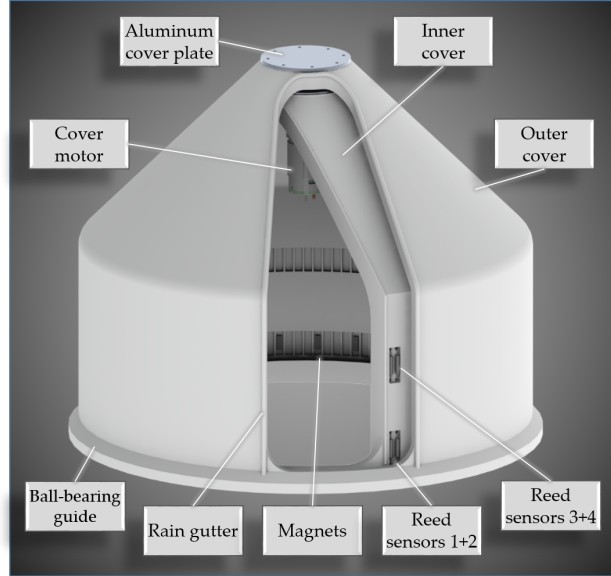

**Figure 3.** CAD model of the newly designed cover (outer + inner one) with a small opening and a steeper slope compared to the first version in Heinle and Chen (2018). With the help of the reed sensors 1+2, the relative position of the cover is calculated (in $10°$ steps). The second sensor is indicating the direction. The reed sensors 3+4 are used to determine the absolute zero position each time before the cover is opening.

the rain sensor data, powering the spectrometer and monitoring the UPS. For the new version, we separated these safety operations from the measurement-related software that is running on a Windows computer to make these features fail-safe. As the microcontroller is in the new version replaced by a PLC, the ECon software needed to be renewed as well.

ECon is structured as a sequence control that loops through the main program, which is grouped into several functions, over and over again. These functions include for example the detection of any alarm caused by the UPS, encoder or power failures, the request of the current solar azimuth angle and the control of the cover motor and other outputs such as relays or signal lamps.

The most safety-relevant function is thereby the control of the cover motor. The program is structured in a way that closing the cover is prioritized in any condition. Even in case of a Reed sensor failure, the program will make sure that the cover closes correctly by evaluating the sensor signals, which are implemented redundantly.

Furthermore, ECon monitors whether the ethernet connections to the computer, spectrometer and internet are working properly. In case that any malfunction is detected, the program automatically restarts either the spectrometer, computer or the router depending on the kind of failure by shortly interrupting the power supply of the respective device using relays. This approach ensures a minimum requirement for human interactions in case of malfunctions, which is particularly beneficial for operating very remote sites.

To keep the temperature within a predefined range, ECon also controls the temperature inside the enclosure by either powering the heater or the fan depending on the actual and the given nominal temperature.





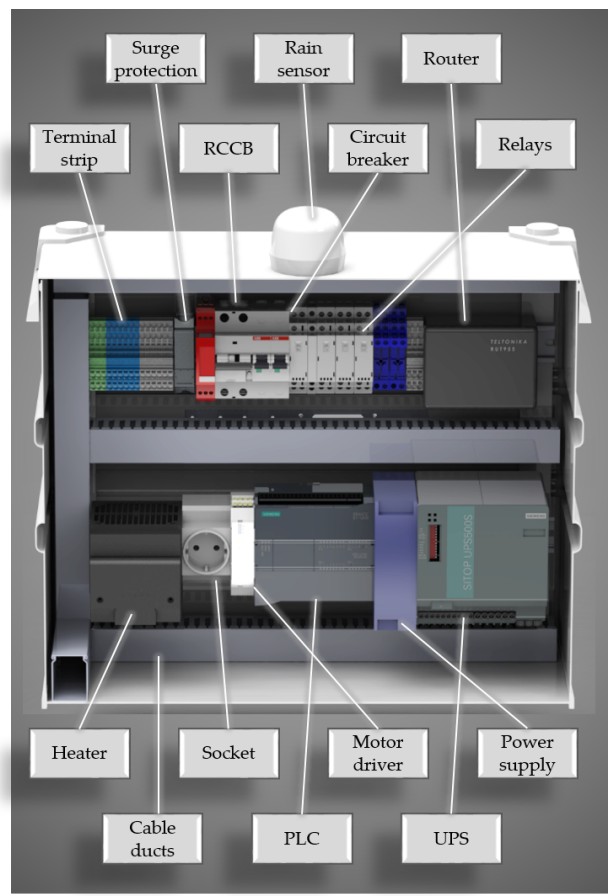

**Figure 4.** CAD model of the electrical components inside the enclosure.

### 3.2.2 Automation software (Pyra)

In order to control the measurements of the spectrometers automatically, it was necessary to develop a software that covers all the tasks that a human operator normally does to perform the measurements. We decided to use Python as a programming language to develop both the automation software and a user interface that allows an operator to set all necessary parameters and observe the current state of the system. The program is running all the time on each enclosure computer and serves as a juncture between the spectrometer, the enclosure system and the operator. As the measurements are based on the spectral analysis of the sun, the program is called Pyra, which is a combination of Python and the name of the Egyptian sun god Ra.

The manufacturer Bruker provides the EM27/SUN spectrometers with the two software OPUS and CamTracker to control both the spectrometer itself and the camera-based solar tracker that is attached to the spectrometer. Pyra does not replace these two software but provides the possibility to start, stop and control them automatically. Besides these necessary tasks, Pyra also monitors the operating system and the spectrometer to detect malfunctions such as insufficient disk space or non-working



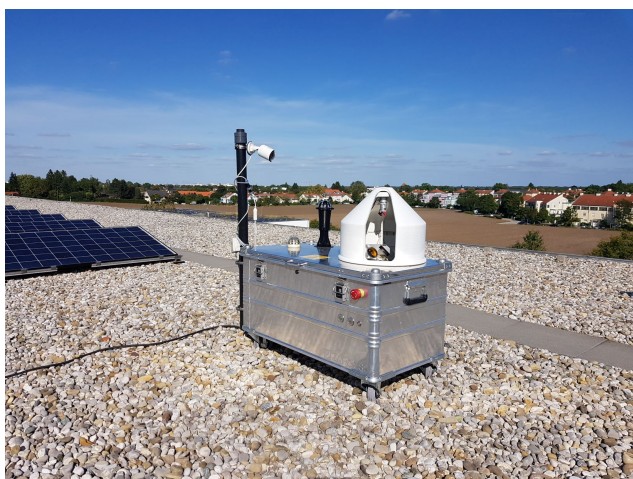

**Figure 5.** Image of the new enclosure system on the roof of a school at our southern site Taufkirchen. The systems includes, inter alia, the newly designed rotating cover, the lightweight aluminum case, the solar radiation sensor and a surveillance camera attached to a post.

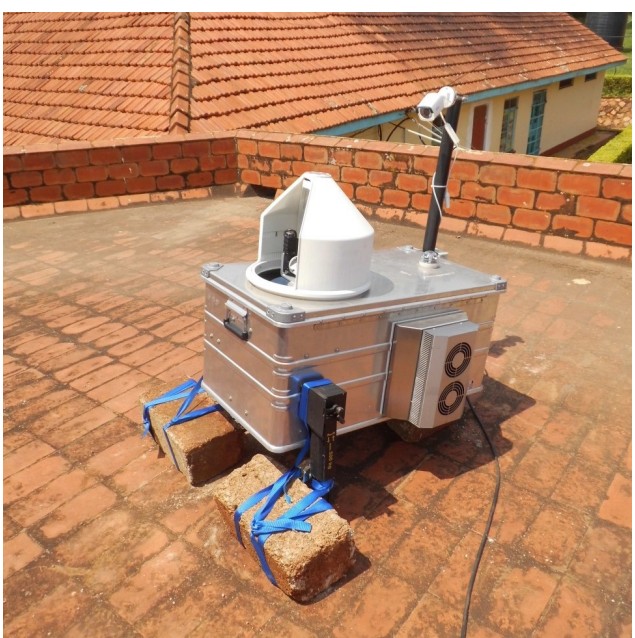

**Figure 6.** Setup of the tropical version of our enclosure in Jinja, Uganda (Latitude: $0.4°$ N). With the help of car jacks the whole system is tilted in order to enable measurements at high elevation levels close to $90°$. Furthermore, the system is equipped with two $150\,\mathrm{W}$ thermo-electrical coolers (attached at the two sides of the system) that keep the temperature inside the enclosure constant at $25\,°\mathrm{C}$.

Photo by Neil Humpage, University of Leicester.





connections. Furthermore, it evaluates whether the environmental conditions are suited for measurements and logs each event to a file.

There are four different operating modes of Pyra. The manual one, where the operator can start and stop the measurements
with just one click, two semi-automated modes, where Pyra is starting and stopping the measurements based on either a defined time or the solar zenith angle (SZA) range and the fully-automated mode. In the latter, Pyra is evaluating the direct solar radiation sensor data and combines it with the online calculated SZA information to start and stop the measurements whenever the environmental conditions are suited.

A more detailed description about the features of the Pyra can be found in Appendix A.

Although, Pyra was developed to automate the process of EM27/SUN spectrometers that are operated in our enclosure system, it can also be used without this system or in a different shelter. In this case, only the fully-automated mode is not working anymore as the information of the direct solar radiation sensor is not available. However, all the other modes are working, which leads to less human effort and more reliable measurements.

All the aforementioned features of Pyra are combined in a common user interface (see Fig. 7). It is a clear and handy
interface that allows any operator to do all the necessary settings for performing automated measurements using EM27/SUN spectrometers. There are in total three Pyra tabs (Measurement, Configuration and Log) and one ECon tab, which we also included in this user interface. The ECon tab allows us to control the PLC that operates the enclosure system (details see section 3.2.1). Thereby, the program itself is not running on the enclosure computer but on the PLC, which makes the safety-related features fail-safe. As the PLC does not provide an own graphical user interface, we decided to include these functions
such as controlling the cover motor, heater, fan, relays, etc. in the Pyra UI as well. For that, the Python library *snap7* is used, which allows to communicate with a Siemens S7 PLC using an ethernet connection.

### 3.2.3 Automated retrieval process

For a fully-automated greenhouse gas observation network, not only the measurements need to be autonomous but also the data processing. Therefore, we automated the data processing chain as well.

Each enclosure computer automatically uploads at the end of a measurement day all the interferograms and the weather data over an SSH connection to our Linux cloud server at the Leibniz Supercomputing Center in Garching. After the data of a full day from a station is completely transferred, the retrieval algorithm that is based on the software GFIT (Wunch et al., 2011; Hedelius et al., 2016) is retrieving the concentration data from the interferograms and stores all values in an ASCII file. This file contains the concentrations as well as the wind information throughout the day.

## 4 Network setup

We tested the automated network consisting of five spectrometers in a measurement campaign in August 2018 (Dietrich et al., 2019), before the permanent network was installed in September 2019. In addition, our first enclosure system is measuring on the university rooftop since 2016 permanently.





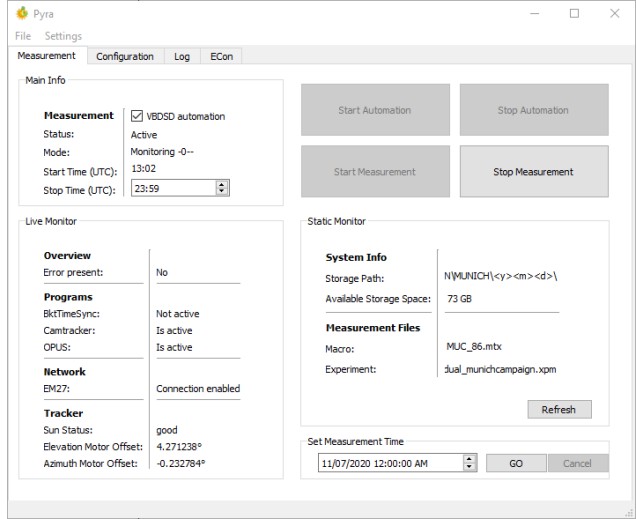

**Figure 7.** User interface of the control software Pyra. In total there are four different tabs (*Measurement, Configuration, Log and Enclosure control*) that can be selected. In this image, the *Measurement* tab is shown.

## 4.1 Test campaign - Munich August 2018

After building in total five enclosure systems, we established the world's first fully-automated GHG sensor network based on the differential column measurements (DCM) principle (Chen et al., 2016) in a one-month measurement campaign in Munich.

For testing our enclosure systems and the network configuration, we borrowed spectrometers from the Karlsruhe Institute of Technology (KIT) and the German Aerospace Center (DLR). Besides our long-term operating station in the inner city, we set up a system in each compass direction, respectively (see red shaded enclosure systems in Fig. 8). The distance between

the downtown station and the outer stations was chosen as approximately 20 km each to ensure that the outer stations are not directly affected by the city emissions if they are located upwind.

Thanks to the automation, we were able to measure on each of the 25 sunny August days mostly from the very early morning to the late evening (approx. 07:00 to 20:00 local time) both on weekdays and weekend days. Thereby, the human interactions were reduced to a minimum and were mostly restricted to setting up and disassembling the enclosure systems on the rooftops

that we used as measurement sites. Therefore, this campaign was characterized by a very small effort as well as a very high data amount. These results are the desired outcomes of such campaigns and are the foundation to use such a setup also for a permanent urban GHG observation network.

## 4.2 Permanent Munich GHG network setup

Although the configuration of the outer stations in the August 2018 campaign was well suited to capture the background

concentrations, this setup cannot be used to determine the pure emissions of the city of Munich. Instead, the greater Munich area emissions are captured as well. As our focus lies in the city emissions itself, we decided to go closer to the city boundaries





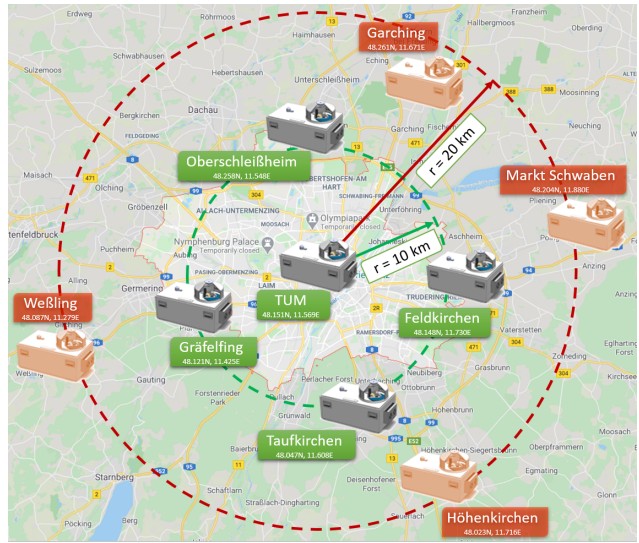

**Figure 8.** Map of the greater Munich area together with the two different sensor network setups that have been implemented. The urban area itself is the highlighted shape in the center. The light red shaded sensor systems together with the center station represent the setup during our summer campaign 2018; the four systems on the green circle together with the center station represent the current setup of the stationary network. Both setups are characterized by having a center station and a station in each compass direction to measure the inflow and outflow of GHG concentrations at arbitrary wind conditions. Map data are from © Google Maps

for our permanent sensor network. The distance between the downtown station and each outer station was halved to 10 km (see green enclosure systems in Fig. 8). Thus, the outer stations are located approximately at the city boundaries of Munich. The second benefit of such a dense sensor setup is that it can be better used for validating concentration gradients measured
by satellites. Due to the unique dataset of our sensor network, the NASA satellites OCO-2 and OCO-2 are measuring $CO_2$ concentrations over Munich in the target mode since spring 2020. The area OCO-2 can cover in this mode over Munich is approximately 21 km × 13 km. As the satellite trajectory is not exactly aligned in the north-south axis, the distance of 10 km between the inner and outer stations is optimal in order to capture the urban concentration gradients.

In addition to the relocation, the enclosure systems were slightly improved based on the experiences from the August 2018
campaign. This includes especially adding a direct solar radiation sensor in order to start and stop the measurements depending on the actual weather conditions. Furthermore, we replaced the three borrowed spectrometers by our own ones so that all five instruments can measure long-term.

All in all, we were able to set up the first permanent urban column concentration network for GHGs in September 2019 using our own five spectrometers. Starting from this date, we are not only measuring the absolute GHG concentration trend of
Munich but also the city gradients, which will be used to determine the urban GHG emissions of Munich over the course of the years and to find unknown emission sources.



## 5    Results

Since 2015, we are continuously measuring the GHG concentrations in Munich with at least one instrument. Over time, the data amount is increasing significantly as we improved our automation and used more and more instruments.

### 5.1    Seasonal Cycle

In Fig. 9 we show the measurement curve of our downtown station over the first five years of measurements. In order to display the seasonal cycle, a sinusoidal function of the form

$$c_{CO_2}(t) = a \cdot (sin(\frac{2\pi(t-b)}{365})) + ct + d \tag{1}$$

with the parameters a to d is fitted. One can see clearly the globally rising trend in $CO_2$ (about $2.4\,\mathrm{ppm}$ per year) as well as the seasonal cycle for the five years.

Although the whole time period between fall 2015 and summer 2020 is covered, there are times with a much larger amount of data compared to the rest. These data hot spots are representing our campaigns in summer 2017 and 2018 as well as our Oktoberfest campaign 2018. A further hot spot can be detected in fall 2016 where the first version of our enclosure system (Heinle and Chen, 2018) has been established and intensively tested in the semi-automated mode. Since summer 2019 the fully-automated enclosure system is measuring whenever the weather conditions are suited, which results in a very high and dense data amount.

In total, we have measured on 498 days throughout the last five years. Thereby only days with continuous measurements of at least 1 hour are taken into account. The ratio of measurement days compared to non-measurement days is about $17\,\%$ for the time period before summer 2019. After establishing the full automation, this ratio increased to about $52\,\%$, which shows the great benefit of our fully-automated sensor network approach.

### 5.2    Side-by-side calibration and gradient comparison

The results in the previous section show that our automation is working and that we are able to gather a lot of GHG measurement data. The final goal of our network is, however, the quantification of the urban emissions. For that, the gradients between the single stations need to be analyzed. As the concentration enhancements of column averaged dry air mole fractions are quite small for an urban emission source, it is absolutely necessary to calibrate the instruments regularly. Besides calibrating the absolute concentrations values next to a TCCON station from time to time, the relative comparison between the single instrument is even more decisive. Therefore, we calibrate all instruments regularly twice a year and additionally at the beginning and end of each campaign. The setup of such a side-by-side calibration measurement day is shown in Fig. 10, where five automated sensor systems are measuring next to each other on top of our university roof. For each instrument and gas species a constant calibration factor is determined using linear regression with $R^2 > 0.9$. As absolute reference value we take the concentration value of the instrument that was last calibrated at a TCCON station.



Fig. 11 shows the $CH_4$ gradients of a standard measurement day on a Saturday during Oktoberfest 2019. It indicates that our sensor network can detect the differences in $CH_4$ concentrations well, which allows us to determine the urban emissions using these measurements as an input. Furthermore, one can see that our automated network allows us to measure not only on weekdays but also on weekends from early morning to evening without the need for human resources.

In Fig. 12 we show the $CO_2$ concentration enhancements above the background concentration for the four outer city stations depending on the wind direction. For this purpose, we use an ultrasonic wind sensor (Gill WindObserver II) on a roof in the inner city of Munich (48.148° N, 11.573° E, 24 m agl.). To determine the background concentration, we use the data of all our measurement stations and determine the lowest measurement point for each time step. Afterwards a moving average with a window size of 4 h is used to smooth the curve as we assume that the background concentration must not change rapidly. For each station a polar histogram shows where and how often the concentration enhancements come from. In contrast to a standard wind rose, the different colors indicate the strength of the concentration enhancement; yellow means low and red high enhancement. The wind speed is displayed by the distance of the respective cell to the center point of each circle.

One can see clearly that for all 4 stations the enhancements are higher towards the city. For the eastern station for example the highest enhancements, indicated by the reddish color, are located in the west. These results indicate that the gas molecules are mainly generated in the city and that our network is able to detect and quantify such urban emitters. Due to technical issues, the southern station started its measurements a bit later than the other four stations. Therefore, the data amount is a little bit less. This is, however, not decisive as north wind in Munich is quite rare.

## 5.3 Influences of the COVID-19 lockdown

Thanks to the automation, we measured throughout the COVID-19 lockdown in spring 2020, which results in a unique dataset showing the influences of such a drastic event to the GHG emissions of a city like Munich. Fig. 13 displays the gradients between the measurements of the inner city station and the background concentrations (cf. section 5.2).

All concentration enhancements are clustered into biweekly bins. In Fig. 13 the median of these bins is displayed as the blue curve. The error bars indicate the $1\sigma$ standard deviation of these biweekly distributions. In addition, the Munich traffic amount is displayed in red using the congestion rate that is provided by TomTom International BV. Furthermore, the COVID-19 lockdown period is shown as the grey shaded area.

The plot demonstrates that the lockdown had a large impact on the traffic amount and that there is a good correlation with the $CO_2$ concentration enhancements. Both curves decrease drastically at the beginning of the lockdown and increases again after the strict restrictions were loosened bit by bit. These results prove that our network can detect changes in the urban emissions well.

Nevertheless, our statistical approach, which uses about 100k measurement points, shows large variations in the $CO_2$ enhancements for the single bins. Such high variations are, however, not concerning as the approach is not considering wind speed and direction for example. Furthermore, the assumption of homogeneously distributed emissions sources does not reflect the truth and photosynthetic effects are not considered. Therefore, it can only serve as a first indication of how the emissions were





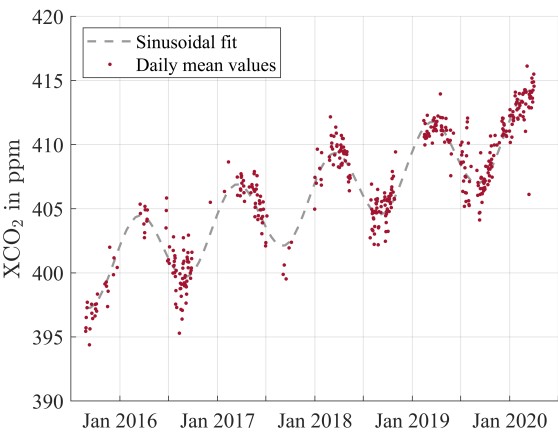

**Figure 9.** Daily mean values of the $CO_2$ measurements from the downtown station in Munich. The concentrations are following the globally rising trend. Furthermore, the seasonal cycle with lower concentrations in summer and higher concentrations in winter is clearly visible for the shown five years.

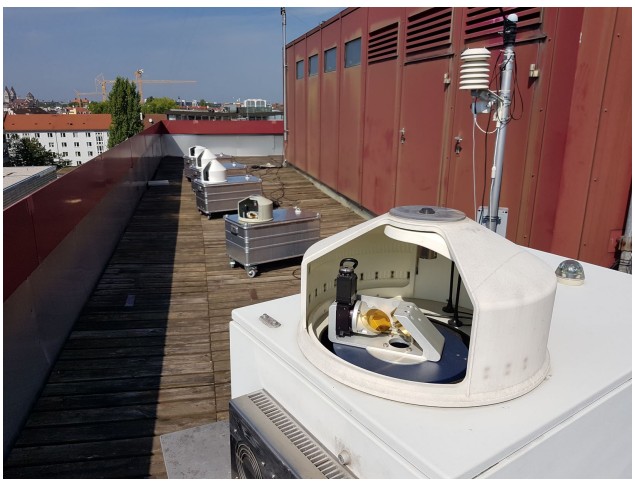

**Figure 10.** Calibration measurements of all our five sensor systems on top of our institute's building. One can see four slightly different versions of our enclosure systems.

reduced during the lockdown period. For the future, we will apply more sophisticated modelling approaches to quantify the emissions.




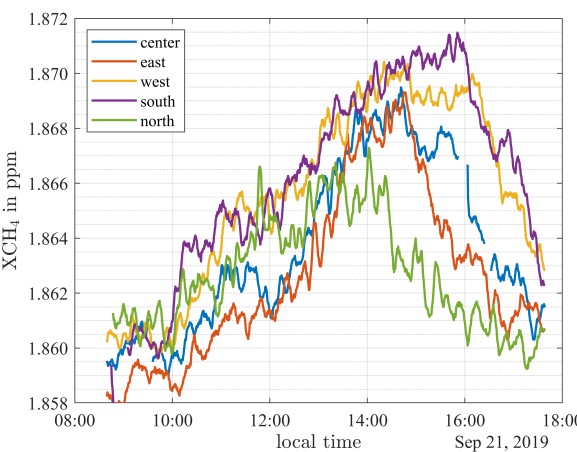

**Figure 11.** CH$_4$ measurement values (5-minute average) of all five stations during our Oktoberfest 2019 campaign on September 21, 2019. The concentration gradients between the single stations are clearly visible, which indicates the presence of strong CH$_4$ sources in the city.

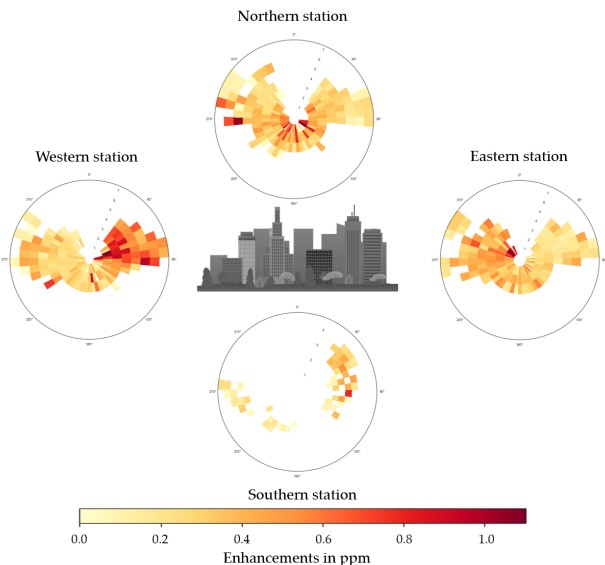

**Figure 12.** Concentration enhancements over the background for each of the 4 outer stations displayed as a polar histogram. The CO$_2$ enhancements are represented by the different colors (low=yellow to red=high). The wind direction is indicated by the location of the respective cells in the circle and the wind speed by the distance of the cells to the center point.

## 6 Conclusion

We present the world's first permanent urban GHG column network consisting of 5 compact solar-tracking spectrometer systems distributed in and around Munich. We developed the hardware and software to establish such a fully automated GHG





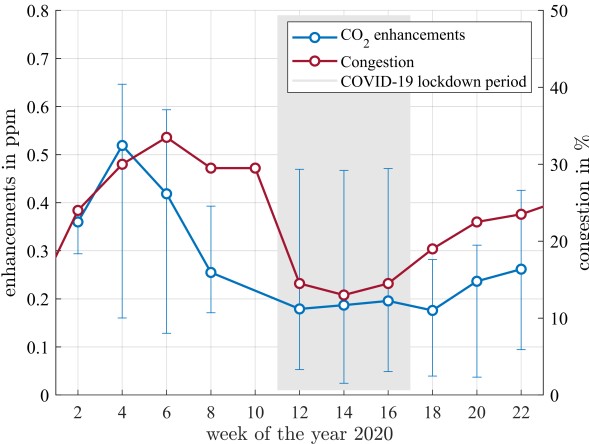

**Figure 13.** Correlations between the $CO_2$ enhancements over the background measured by our inner city station in Munich and the traffic amount represented by the congestion rate. The time period includes the COVID-19 lockdown in spring 2020. We show the median trend of all column concentration gradients clustered into biweekly bins. The error bars indicate the $1\sigma$ standard deviation of these enhancements. Traffic data are from ©2020 TomTom International BV.

sensor network for quantifying large area emission sources such as cities. Both the enclosure system and the related Python program for automating the measurement process can be used by the community to build up similar sensor networks in cities worldwide. Also, COCCON would significantly benefit from such an automated system, as the current approach of operating EM27/SUN spectrometers in this network still requires manpower on site for starting up measurements and for protecting the spectrometer from adverse meteorological conditions. Permanent and long-term observations will help to improve the
understanding of the global carbon cycle.

With our sensor systems, we did several test campaigns between 2016 and 2019 and finally set up the world's first permanent urban GHG sensor network based on the differential column methodology in fall 2019. The results show the advantages of such an automated network such as very high data amount, low personnel effort and high data quality. Due to the very frequent measurements that took place independent on the day of the week or the season, we show in this study that our network can
detect both the globally rising trend of $CO_2$ concentrations and the seasonal cycle very well.

The final goal of such a network is the quantification of urban GHG emissions. For that, the concentration gradients between the downwind and upwind stations are decisive as they are representing the amount of pollutants that are generated by the emission sources in between. Our results demonstrate that these gradients can be captured clearly with our sensor setup. Additional analyses including wind information demonstrate that the city is causing these emissions.
Furthermore, the network can be used to validate GHG satellites in a unique way as not only absolute values but also concentration gradients can be compared. Since spring 2020, the NASA OCO-2 and OCO-3 satellites (Crisp, 2016; Eldering et al., 2018) have been measuring urban $CO_2$ concentration gradients of Munich using the spatially highly resolved target mode in a recurring pattern to compare the satellite measurements with our ground-based ones.



Thanks to the full automation, we were able to measure the concentration gradients also during the COVID-19 lockdown
period in spring 2020. The results show a clear correlation between the $CO_2$ column concentration gradients and the traffic
emissions, which were both drastically influences by the lockdown.

In order to quantify the Munich GHG emissions, we are currently developing an atmospheric transport model based on
Bayesian inversion. Such a modeling framework will help us in the future to quantify the GHG emissions of Munich and find
correlations between parameters such as time of the day, season, weather conditions, etc. Furthermore, we will use our rich
dataset to detect and quantify unknown GHG emission sources.

In sum, this study provides the framework for establishing a permanent GHG sensor network to determine urban emissions
using column measurements in any city worldwide. The characteristics of the presented approach, such as high precision, ease
of use and low operating costs, form the basis for it to become a new standard for monitoring urban GHG emissions.



*Code and data availability.* The Python software Pyra as well as the technical drawings, schematics and component list for the enclosure
system can be provided by the authors upon request.

## Appendix A: Pyra - software features

To control the spectrometer program OPUS, we use the Microsoft Windows technology *dynamic data exchange* (DDE), which
is also supported by OPUS. It is a protocol to exchange data based on the client-server model and allows us to send requests,
such as starting a measurement or loading a specific setting file, to OPUS. With the help of DDE, combined with an MTX
macro file for OPUS, Pyra can start recurring measurements of the spectrometer. The necessary settings are stored in an XPM
experiment file and are loaded into the program in the same way.

The communication with the solar tracker program CamTracker is done in a simpler way as the settings of this program
do not need to be changed after the initialization anymore. Therefore, we asked the manufacturer Bruker to implement an
auto start option for the tracker. Whenever CamTracker is called with this option, the solar tracker automatically aligns its two
mirrors to the calculated live position of the sun and enables the tracking of the sun. After terminating the program, the tracker
automatically moves back to its parking position.

In order to detect malfunctions, Pyra is equipped with several live monitoring functions. It monitors every $0.2\,s$ whether the
two programs OPUS and CamTracker are still running correctly. If this is not the case anymore, it automatically restarts the
non-working program to proceed the measurements. Furthermore, the log files of CamTracker are read continuously, which
allows us for example to automatically detect if the solar tracker is not tracking the sun correctly anymore. Such a behaviour
is quite common as the solar tracker is using a camera-based approach to follow the sun in the course of the day. In case of
cloudy conditions, the algorithm sometimes mistakenly detects objects other than the sun, resulting in an incorrect tracking. In
such a case, the tracking is restarted using the calculated position of the sun at the given coordinates and time. In addition to
trying to solve the error automatically, Pyra is also sending an error notification email to an operator, whose email address can
be defined in the settings.

*Author contributions.* FD and JC conceived the study and developed the concept, FD led the hard- and software development as well as the
setup of the sensor network. FD, BV and BR built the enclosure systems. PA, BV and FD developed the software Pyra. FD and JC have
performed the measurements. FD, JC and NN analyzed the measurement data. FD and JC wrote the manuscript.

*Competing interests.* The authors declare that they have no conflict of interest.

*Acknowledgements.* We thank Ludwig Heinle for developing the first version of a semi-automated enclosure system; Frank Hase for testing
and calibrating the instruments prior to the delivery and for providing us two spectrometers each for our August 2018 and Oktoberfest





2019 campaign; André Butz for providing us his EM27/SUN in our August 2018 campaign; Jacob Hedelius for his support in all matters concerning the GFIT retrieval algorithm; Stephan Hachinger for helping us regarding the automated retrieval process on the Linux cloud; Jonathan Franklin, Taylor Jones, Andreas Luther and Ralph Kleinscheck for their support during our measurement campaigns; Neil Humpage

and Harmut Boesch for testing our enclosure system in Uganda; Martin Wild, Norbert Tuschl, Abdurahim Bingöl, Sebastian Zunterer and Bernhard Obermaier for manufacturing the enclosure systems; Markus Garhammer and Mark Wenig for providing us meteorological data; the municipalities of Feldkirchen, Gräfelfing, Mark Schwaben, Oberschleißheim and Taufkirchen as well as the ARCONE Technology Center Höhenkirchen who have provided us their rooftops as measurement sites; and our students Andreas Forstmaier, Adrian Wenzel, Jared Matzke, Yiming Zhao, Xu Hang, Dingcong Lu, Xiao Bi and Michal Wedrat for their help during the campaigns and the network setup as

well as programming helpful automation scripts and supporting the CAD model.

The project was funded by the Deutsche Forschungsgemeinschaft (DFG, German Research Foundation) – CH 1792/2-1; INST 95/1544. Jia Chen is supported by Technical University of Munich – Institute for Advanced Study, funded by the German Excellence Initiative and the European Union Seventh Framework Program under grant agreement number 291763.



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
