# Peer review of "Munich permanent urban greenhouse gas column observing network"

_Atmospheric Measurement Techniques, 2020_

## Referee Comment (RC1) · David Griffith (Referee) · 22 Sep 2020

This paper describes a measurement system and early results from a set of portable, automated ground-based solar infrared spectrometers based on the commercially-available Bruker EM27-Sun. Each spectrometer is housed in an autonomous housing allowing weather-proof operation and full automation. Five such spectrometers are deployed around Munich, Germany to deduce city-scale emissions of CO2, CH4 and CO from the upwind-downwind differences of total column amounts of these gases. The paper provides full technical details, building on an earlier 2106 publication. It represents a substantial instrumental development which would be of interest to anyone concerned with quantifying extended-source emissions such as cities or large point sources. It would require substantial effort and funding for others to duplicate the work,

so I am pleased to see that the authors agree to make all technical plans, drawings and code available. It does leave me curious whether the authors have any plan to licence the system to a commercial provider – weather protection and automation for solar trackers and the EM27 used here and in TCCON and COCCON networks are not commercially available, but every installation needs one. The paper is clearly suitable for publication in AMT, after minor revisions and extensions and a few technical corrections.

General comments

Section 5.2 describes essential side by side comparisons (they are not "calibrations" per se) amongst the 5 instruments and TCCON FTS at Karlsruhe. These comparisons are of critical importance in evaluating small differences between upwind and downwind measurements, since any instrument bias would be interpreted as a gradient if not corrected. Yet no details of the comparisons are presented, and the reader has no idea of the uncertainty in the bias correction factors. It is essential to present the numerical details from all 6-monthly comparisons between instruments (and occasional TCCON comparisons). This could be done as a table of regression factors for each instrument pair and date. Only then can the reader assess the statistics of these comparisons – their magnitude, stability and reproducibility. The quantitative uncertainty will be essential for later modelling of the measured gradients in any Bayesian inversion scheme.

In section 5.3 and Figure 13 upwind-downwind data are compared, and at L302 the changes during Covid lockdown in 2020 are compared. However without the quantitative intercomparison data requested for section 5.2, it is impossible to assess the meaningfulness of these differences. How are the error bars on the $CO_2$ enhancements calculated (refer to 5.2 discussion)? I also do not agree that the data show a correlated drop in $CO_2$ enhancement with traffic congestion from weeks 4 to 12. $CO_2$ enhancement drop 4-5 weeks earlier, meanwhile the season is changing from winter to spring and presumably $CO_2$ sources other than traffic also change in this time, such

as home, industrial and commercial heating. The interpretation is too simplistic. Thus I do not agree with the statement "These results prove that our network can detect changes in the urban emissions" – see also the conclusion around L 335. Based on the detail currently presented this conclusion is not valid. However this is not to say it is not possible. The data should be extremely valuable for such interpretations when combined with a city-scale regional model such as described as being under development. I agree that this level of modelling and interpretation is outside the scope of this paper.

Finally, I count the term "world's first" 5 times in the manuscript – this is excessive. I suggest it is OK and sufficient to point his out once-only in the abstract, conclusion, and introduction.

Technical corrections

L15: Suggest rewording " as well as concentration gradients between sites upwind and downwind of the city."

L59: TCCON measurements are made at a resolution of 0.02 cm-1

L92: "Reed sensors in the inner cover COUNT these signals..."

L93: "Reed sensors INDICATE the absolute position." There are several more cases of present perfect tense where simple present is normal English usage – a copy editor should pick these up.

L119: As described in the current text, the system could not operate for several hours around noon in the southern hemisphere unless the whole instrument is rotated 180° to point north, with software able to handle this switch. Presumably this is the case - I suggest the text be clarified to make this clear. Further, why is the range (30-300°) not symmetric around North?

L138: Unclear wording ("cannot" should be "can not"), I suggest either that small change or "They can control temperature to a constant level... as well as condense

(not condensate) water vapour . . ."

L178: . . . with two INDEPENDENT software COMPONENTS, OPUS and Camtracker, to control. . ..

L207: Very little detail of the spectrum retrieval with GFIT is given. This is OK if it follows the Wunch and Hedelius references exactly, but any variations from those procedures should be described because they will impact on accuracy and precision. In particular, when is the analysis done? – vertical pressure-temperature-humidty profiles only become available after a few days, but the text sort-of implies the fitting is done the same day in a pipeline process.

L219: "respectively" is not needed here, remove.

L230: "pure emissions" is not quite the right wording, I suggest ".. this setup cannot be used to determine the emissions of the central city of Munich separate from its outer surrounds".

Fig 8 L2: The urban area itself is largely contained (provide a %?) within the green inner dotted circle in the centre.

L236: OCO-2 and OCO-3

L254: with the parameters a to d to be fitted.

L270: See general comments, this section should be expanded to include actual regression coefficient and statistics.

L292: Can you provide the actual starting dates and numbers of measurements, rather than "a little bit later" and "a little bit less".

L302: See general comments.
* * *

---

## Referee Comment (RC2) · Anonymous Referee #2 · 4 Nov 2020

General comments:

In this study, Dietrich et al. report on a novel permanent urban greenhouse gas monitoring network using EM27/SUN spectrometers inside an automated enclosure system in Munich. They carefully describe the technical innovations from a previous design as well as results from a successful testing campaign and long-term operations. It is clear that the presented systems are a significant improvement and hold the potential to facilitate such measurements in many cities and regions in the future. The paper is clearly structured and very well written and it fits perfectly into the scope of AMT. Although the technical aspects are overall excellent, there is unfortunately a major point of concern that should be addressed before publication. The authors have made very strong statements that the manuscript itself does not address. For example, the claim

that the presented systems and approach allows to determine urban greenhouse gas emissions 'in any city worldwide'. More instances of such sweeping statements are given in the specific comments section. I recommend that the authors revisit these statement and provide additional data and explanations to support them. On the other hand the author could also choose to let the fully supported and very impressive results, e.g. increased data availability, continuous operations during COVID-lockdown, tracking of XCO2 enhancement changes speak for themselves.

Specific comments:

L1 – Consider removing 'the'

L8 – This study does not establish that this technique by itself allows to quantify emissions. For example, how well can annual emissions be estimated when observations have a clear-sky (and maybe seasonal) bias.

L21 – Although it is an impressive measurement system for total column CO2 and CH4, it seems far from proven that this technique and system as a 'new standard for determining GHG emissions', given the complexity and challenges in urban environments.

L27 – Gurney et al. did not claim that urban areas contribute more than 70% of GHG emissions, but rather that 'Cities account for more than 70% of global fossil fuel emissions'. There are other non-urban and non-fossil fuel sources that contribute significantly to global GHG emissions, like land-use and land-use change (CO2), agriculture (CH4, N2O), etc. Please correct this statement or provide a reference for your claim.

L59 – Do all TCCON stations use this very high spectral resolution in their operations?

L64 – How can you be sure that you will be able to assess the effectiveness of mitigation strategies? Could the atmospheric modelling framework not be insufficient to achieve this, if for example urban heat island effects are not correctly modelled. Furthermore, are the planned emission reductions in Munich large enough to significantly alter XCO2, XCH4, XN2O and other greenhouse gases.

L147- Consider rephrasing for readability

L177 - See L147

L219 - consider removing 'respectively'

L230 - What is meant by 'pure emissions'? Does this refer to net emissions of the city of Munich?

L236 – 'OCO-2' is repeated here

L252 (eq1) - Why was such a simplified fitting approach taken here, when more suitable and well-established methods are widely used to determine seasonal variations and trends in atmospheric CO2 records? For example, as described in Nakazawa et al. 1997 and references therein (https://doi.org/10.1002/(SICI)1099-095X(199705)8:3<197::AID-ENV248>3.0.CO;2-C).

L257 - The word 'hotspot' seems not to be optimal to describe data density

L264 - It would be worthwhile to explain if this refers to 52% of all days since automation or all sunny/suitable days since automation, in any case a very impressive result.

L281 - Adding the pollution rose plot for CO2 enhancement of the station inside the city could also be very interesting here to learn about the source distribution inside the city limits.

L293 – How much less data is available for southern station

L296 - This study does NOT show the drastic impact on GHG emissions, but mere a decrease in local GHG enhancements. There are many other possible reasons for changes in GHG concentrations other then emission changes. It is reasonable to assume here that the concentration enhancement change is due to emission changes, but this should be stated carefully and other potential sources of uncertainty have to be included when referring to emissions.

L302 - Please provide the R2 for this relationship. Also, looking at figure 14 it seems clear that $CO_2$ enhancements decreased strongly in week 6 and 8 already, well before the lockdown period, while congestion was above 25%, i.e. fairly normal. A scatter plot of the two quantities could be a useful addition in the supplemental information of this paper.

L304 - See comment L296, L302, this study does not establish a decrease in emissions within Munich. Further modelling (including biospheric $CO_2$) and assessment of uncertainties seems necessary before the authors should claim that they have proven that their system is sufficient to track emission changes. The authors later refer these uncertainties, so they seem aware of this problem, so why make such a strong claim here? Being able to reliably track XCO2 enhancement changes during COVID lockdown with an automated system is already an excellent achievement in itself.

L327 - This statement completely ignores the potentially large impact on $CO_2$ concentrations by the urban biosphere, that has been found to be an important $CO_2$ sink (and sometimes source) in urban areas, for example, Miller et al. 2020 (PNAS, https://doi.org/10.1073/pnas.2005253117).

L335: No data set of traffic emissions was presented in this paper. I agree that the seen decrease in congestion makes emission reductions extremely likely, but this should be stated carefully. Also the decrease does not seem to be concurrent.

L342: It is unclear how this study has established that column measurements can be used in 'any city worldwide. It seems apparent that the concentration gradients in the total column for smaller cities might be too small to detect reliably or the $CO_2$ emission signal might be masked due to biospheric uptake in cities in the tropics. What about cities with very strong aerosol loads, like Beijing, would the EM27SUN be able to penetrate dense smog?

---

## Author Comment (AC1) · 2 Dec 2020

**1   Responses to the comments of reviewer 1**

We would like to thank David Griffith for thoroughly reading our paper and providing very helpful and insightful comments.  Below, please find our responses to his comments.

[Figure]

**2 Summary**

**Reviewer:** This paper describes a measurement system and early results from a set of portable, automated ground-based solar infrared spectrometers based on the commercially available Bruker EM27-Sun. Each spectrometer is housed in an autonomous housing allowing weather-proof operation and full automation. Five such spectrometers are deployed around Munich, Germany to deduce city-scale emissions of $CO_2$, $CH_4$ and CO from the upwind-downwind differences of total column amounts of these gases. The paper provides full technical details, building on an earlier 2016 publication. It represents a substantial instrumental development which would be of interest to anyone concerned with quantifying extended-source emissions such as cities or large point sources. It would require substantial effort and funding for others to duplicate the work, so I am pleased to see that the authors agree to make all technical plans, drawings and code available. It does leave me curious whether the authors have any plan to licence the system to a commercial provider – weather protection and automation for solar trackers and the EM27 used here and in TCCON and COCCON networks are not commercially available, but every installation needs one. The paper is clearly suitable for publication in AMT, after minor revisions and extensions and a few technical corrections.

**Response:** Thank you very much for appreciating our work and supporting its publication in AMT. Thank you also for suggesting the commercialization of our devices. We are currently looking into the possibilities to make our systems commercially available.

**3 General comments**

**Reviewer:** Section 5.2 describes essential side by side comparisons (they are not "calibrations" per se) amongst the 5 instruments and TCCON FTS at Karlsruhe. These

comparisons are of critical importance in evaluating small differences between upwind and downwind measurements, since any instrument bias would be interpreted as a gradient if not corrected. Yet no details of the comparisons are presented, and the reader has no idea of the uncertainty in the bias correction factors. It is essential to present the numerical details from all 6-monthly comparisons between instruments (and occasional TCCON comparisons). This could be done as a table of regression factors for each instrument pair and date. Only then can the reader assess the statistics of these comparisons – their magnitude, stability and reproducibility. The quantitative uncertainty will be essential for later modelling of the measured gradients in any Bayesian inversion scheme.

**Response:** Thank you for pointing out the missing quantitative comparison. We added two new tables (Table B1 and Table B2), which include the calibration factors for both the comparisons to the Karlsruhe instrument, which is calibrated to the TCCON standard, and the comparisons amongst our five instruments.

**Reviewer:** In section 5.3 and Figure 13 upwind-downwind data are compared, and at L302 the changes during Covid lockdown in 2020 are compared. However without the quantitative intercomparison data requested for section 5.2, it is impossible to assess the meaningfulness of these differences. How are the error bars on the $CO_2$ enhancements calculated (refer to 5.2 discussion)? I also do not agree that the data show a correlated drop in $CO_2$ enhancement with traffic congestion from weeks 4 to 12. $CO_2$ enhancement drop 4-5 weeks earlier, meanwhile the season is changing from winter to spring and presumably $CO_2$ sources other than traffic also change in this time, such as home, industrial and commercial heating. The interpretation is too simplistic. Thus I do not agree with the statement "These results prove that our network can detect changes in the urban emissions" – see also the conclusion around L 335. Based on the detail currently presented this conclusion is not valid. However this is not to say it is not possible. The data should be extremely valuable for such interpretations when

combined with a city-scale regional model such as described as being under development. I agree that this level of modelling and interpretation is outside the scope of this paper.

**Response:** Thank you very much for the valuable insights. After adding the requested information in section 5.2 regarding the accuracy of our measurements, it becomes clearer that the enhancements in Figure 13 represent a real signal. Furthermore, we clarified how the error bars in Figure 13 are generated: "The error bars show the $1\sigma$ standard deviation of all enhancements within the respective two-week period."

In addition, we refined our statement regarding the correlation of our measurements to the traffic data. The new formulation is: "The plot demonstrates that the lockdown had a significant impact on traffic flow. The $CO_2$ enhancements show a similar pattern throughout the first half of the year 2020. Based on the regression plot, there seems to be a correlation between the reduced traffic volume and the lower $CO_2$ enhancements ($R^2$=0.63). Both curves first decrease and then increase again after the strict restrictions were gradually loosened." We also added a regression plot (Figure 13, right) showing the correlation between the $CO_2$ enhancements and the traffic congestion data ($R^2 = 0.63$), to provide evidence for our statement.

In the conclusion, we changed the sentence to: "The results show a *possible* correlation between the $CO_2$ column concentration gradients and the traffic emissions, both of which *appear to be drastically affected* by the lockdown."

**Reviewer:** Finally, I count the term "world's first" 5 times in the manuscript – this is excessive. I suggest it is OK and sufficient to point this out once-only in the abstract, conclusion, and introduction.

**Response:** Thanks for pointing this out. We deleted three of the five occurrences of "world's first".

**4   Technical corrections**

1. L15: Suggest rewording "as well as concentration gradients between sites upwind and downwind of the city."

   **Response:** Thanks, we changed it according to your suggestion.

2. L59: TCCON measurements are made at a resolution of $0.02$ cm$^{-1}$

   **Response:** Sorry for the confusion. You are right, the resolution of TCCON measurements is $0.02$ cm$^{-1}$. We changed it in the paper accordingly.

3. L92: "Reed sensors in the inner cover COUNT these signals..."

   **Response:** We changed "are counting" to "count".

4. L93: "Reed sensors INDICATE the absolute position." There are several more cases of present perfect tense where simple present is normal English usage – a copy editor should pick these up.

   **Response:** Thanks for the hint. We changed "are indicating" to "indicate". Furthermore, we tried to change all occurrences of present progressive to simple present.

5. L119: As described in the current text, the system could not operate for several hours around noon in the southern hemisphere unless the whole instrument is rotated 180° to point north, with software able to handle this switch. Presumably this is the case - I suggest the text be clarified to make this clear. Further, why is the range (30-300°) not symmetric around North?

   **Response:** Thanks for pointing this out. The range is not symmetric around North as the first mirror of the solar tracker is not centered at the rotation axis. Therefore, there is a slightly asymmetric behaviour of the morning and evening azimuth. We changed the text to: "That is why we designed our new cover so

that it can measure solar elevation angles up to about 80° and azimuth angles between 30° and 300° for setups at the northern hemisphere. The asymmetric azimuth angle range is due to the non-centered first mirror of the solar tracker. If the system is used in the southern hemisphere, it must be rotated by 180° and a setting must be changed in the software. These solar angles cover most places in the world."

6. L138: Unclear wording ("cannot" should be "can not"), I suggest either that small change or "They can control temperature to a constant level... as well as condense (not condensate) water vapour..."

**Response:** We changed to sentence according to your suggestions to: "They can control temperature to a constant level of 25 °C under normal weather conditions in Uganda as well as condense water vapour to reduce the relative humidity inside the system."

7. L178: ...with two INDEPENDENT software COMPONENTS, OPUS and Cam-tracker, to control....

**Response:** We changed the formulation according to your suggestion.

8. L207: Very little detail of the spectrum retrieval with GFIT is given. This is OK if it follows the Wunch and Hedelius references exactly, but any variations from those procedures should be described because they will impact on accuracy and precision. In particular, when is the analysis done? – vertical pressure-temperature-humidity profiles only become available after a few days, but the text sort-of implies the fitting is done the same day in a pipeline process.

**Response:** Yes, we follow the GFIT retrieval as described in the references. We added a few additional information and modified the sentence about when we start the retrieval algorithm emphasizing that we have to wait for the vertical pressure profiles as you pointed out: "After about five days, when the a priori vertical pressure profiles from NCEP (National Centers for Environmental Prediction)

are available, the retrieval algorithm converts the information from the interferograms into concentrations. The retrieval algorithm used is GGG2014 (Wunch et al., 2015), which is also used to retrieve all the TCCON data. We applied the standard TCCON parameters, including the air mass independent correction factors (AICFs). The spectral windows for retrieving diverse gas species are slightly modified according to the EGI setup (Hedelius et al., 2015)."

9. L219: "respectively" is not needed here, remove.

   **Response:** We deleted "respectively".

10. L230: "pure emissions" is not quite the right wording, I suggest ".. this setup cannot be used to determine the emissions of the central city of Munich separate from its outer surrounds".

    **Response:** Thanks, we changed the sentence according to your suggestion.

11. Fig 8 L2: The urban area itself is largely contained (provide a %?) within the green inner dotted circle in the centre.

    **Response:** In Figure 8, we introduced the boundary of the urban area (black line). Furthermore, we changed the sentence to: "The urban area itself (indicated by the black border) is largely contained within the inner green dashed circle in the center, [...]"

12. L236: OCO-2 and OCO-3

    **Response:** Thanks for pointing this out. We changed it to "OCO-2 and OCO-3"

13. L254: with the parameters a to d to be fitted.

    **Response:** We changed the sentence to: "with the parameters a to d to be fitted".

14. L270: See general comments, this section should be expanded to include actual regression coefficient and statistics.

**Response:** Thank you. See response to general comments. We added two new tables (Table B1 and B2) that includes scaling factors and regression coefficients.

15. L292: Can you provide the actual starting dates and numbers of measurements, rather than "a little bit later" and "a little bit less".

    **Response:** We added a new table (Table 1) that includes the dates when the respective instruments started to measure in our network as well as the measured data points taken so far.

16. L302: See general comments.

    **Response:** See response to general comments. We changed our interpretation according to your advice.

With best regards,
Florian Dietrich on behalf of all co-authors

––––––––––––––––––––––––––––––––

**Fig. 1.** Map of the greater Munich area together with the two different sensor network setups that have been implemented.

---

## Author Comment (AC2) · 2 Dec 2020

**1 Responses to the comments of reviewer 2**

We would like to thank reviewer 2 for reading our paper in detail and giving helpful comments. Below please find our answers:

**2 General comments**

**Reviewer:** In this study, Dietrich et al. report on a novel permanent urban greenhouse gas monitoring network using EM27/SUN spectrometers inside an automated enclosure system in Munich. They carefully describe the technical innovations from a previous design as well as results from a successful testing campaign and long-term operations. It is clear that the presented systems are a significant improvement and hold the potential to facilitate such measurements in many cities and regions in the future. The paper is clearly structured and very well written and it fits perfectly into the scope of AMT. Although the technical aspects are overall excellent, there is unfortunately a major point of concern that should be addressed before publication. The authors have made very strong statements that the manuscript itself does not address. For example, the claim that the presented systems and approach allows to determine urban greenhouse gas emissions "in any city worldwide". More instances of such sweeping statements are given in the specific comments section. I recommend that the authors revisit these statement and provide additional data and explanations to support them. On the other hand the author could also choose to let the fully supported and very impressive results, e.g. increased data availability, continuous operations during COVID-lockdown, tracking of $XCO_2$ enhancement changes speak for themselves.

**Response:** Thank you very much for your helpful and constructive feedback and your appreciation of our technical achievements. Regarding your critics, we agree with you that some of our statements regarding the emission assessment are too strong and not always supported by data or references. Therefore, we modified these statements throughout the text. Please see our comments in the specific comments section.

**3 Specific comments**

1. L1 – Consider removing "the"

   **Response:** We deleted "the"

2. L8 – This study does not establish that this technique by itself allows to quantify emissions. For example, how well can annual emissions be estimated when observations have a clear-sky (and maybe seasonal) bias.

   **Response:** Thank you for pointing this out. In our opinion, the properties of column measurements such as insensitivity to vertical redistribution of tracer masses and surface fluxes upwind the city, are a very important prerequisite to quantify emissions. In addition, we have the column measurements conducted upwind and downwind of the city, and the possible biases are canceled out by looking at the gradients. We slightly modified the formulation to "These column measurements and column concentration differences are relatively insensitive to vertical redistribution of tracer masses and surface fluxes upwind of the city, making them a suitable input for an inversion framework and, therefore, a well-suited candidate for the quantification of GHG emissions."

3. L21 – Although it is an impressive measurement system for total column $CO_2$ and $CH_4$, it seems far from proven that this technique and system as a "new standard for determining GHG emissions", given the complexity and challenges in urban environments.

   **Response:** You are right, this statement is probably a bit excessive. Therefore, we changed the sentence to: "In summary, our achievements in automating column measurements of GHGs will allow researchers all over the world to establish this approach for long-term greenhouse gas monitoring in urban areas."

4. L27 – Gurney et al. did not claim that urban areas contribute more than 70% of GHG emissions, but rather that "Cities account for more than 70% of global fossil

fuel emissions". There are other non-urban and non-fossil fuel sources that contribute significantly to global GHG emissions, like land-use and land-use change ($CO_2$), agriculture ($CH_4$, $N_2O$), etc. Please correct this statement or provide a reference for your claim.

   **Response:** Thank you for pointing out this mistake. We changed it to "70% of global fossil fuel $CO_2$ emissions [...]"

5. L59 – Do all TCCON stations use this very high spectral resolution in their operations?

   **Response:** You are right, the resolution of TCCON measurements is lower (0.02 $cm^{-1}$). We changed it in the paper accordingly.

6. L64 – How can you be sure that you will be able to assess the effectiveness of mitigation strategies? Could the atmospheric modelling framework not be insufficient to achieve this, if for example urban heat island effects are not correctly modelled. Furthermore, are the planned emission reductions in Munich large enough to significantly alter $XCO_2$, $XCH_4$, $XN_2O$ and other greenhouse gases.

   **Response:** Thank you for this note. We modified the language of this sentence a bit: "The combination of our sensor network with a suitable modeling framework will build the basis for monitoring urban GHG emissions over years, identifying unknown emission sources, validating satellite-based GHG measurements as well as assessing the effectiveness of the current mitigation strategies." The details of the modeling framework will be part of a follow up paper. Up to our knowledge so far, the challenges you mentioned can be solved.

   Furthermore, you are right: the absolute values of the column averaged GHG concentrations will not alter significantly based on the planned emission reductions. However, the emission information in our approach is included in the concentration gradients. To date, we can see a clear concentration gradient, with an estimated bottom-up $CO_2$ emissions of about 5.9 t per Munich citizen and year.

As the reduction goals aim to reduce these emissions by about 50% to 3 t per citizen and year until 2030, we are certain that our instruments will sense the changes in concentration gradients.

7. L147- Consider rephrasing for readability

**Response:** We rephrased the sentence: "For controlling and automating the enclosure system, we developed two independent software: ECon and Pyra. The purpose of Econ is to control all safety and enclosure features that are monitored by the PLC, whereas Pyra is used to control the spectrometer and to automatically perform the measurements. Pyra also includes a user interface (UI) where the operator can set all parameters and observe the current state of the system."

8. L177 - See L147

**Response:** Thank you. We reformulated the sentence: "Since the measurements are based on the spectral analysis of the sun, we have named the program Pyra, which is a combination of the programming language Python and the name of the Egyptian sun god Ra."

9. L219 - consider removing "respectively"

**Response:** We deleted "recpectively".

10. L230 - What is meant by "pure emissions"? Does this refer to net emissions of the city of Munich?

**Response:** We changed the sentence to "[...] this setup cannot be used to determine the emissions of the central city of Munich separately from its outer surroundings.".

11. L236 – "OCO-2" is repeated here

**Response:** Thanks for pointing this out. We changed the second occurence of "OCO-2" to "OCO-3".

12. L252 (eq1) - Why was such a simplified fitting approach taken here, when more suitable and well-established methods are widely used to determine seasonal variations and trends in atmospheric $CO_2$ records? For example, as described in Nakazawa et al. 1997 and references therein (https://doi.org/10.1002/(SICI)1099-095X(199705)8:3<197::AID-ENV248>3.0.CO;2-C).

**Response:** Thank you for suggesting us to use a more sophisticated fitting approach. We agree that our simple method cannot be used for a detailed and quantitative investigation of interannual variability in the $CO_2$ trend. For such purposes, a method as described in Nakazawa et al. (1997) would be necessary. However, we use the fitted curve just as a qualitative comparison and visualization in the plot. The obtained fitting parameters are not used in any further analysis. Therefore, we think that the simple fitting approach is sufficient for this case.

13. L257 - The word "hotspot" seems not to be optimal to describe data density

**Response:** We changed it to "These high density data clusters represent our campaigns [...]"

14. L264 - It would be worthwhile to explain if this refers to 52% of all days since automation or all sunny/suitable days since automation, in any case a very impressive result.

**Response:** Thank you for this comment. The two ratios refer to all days not only to suited days. As a result, we measured on average at least one hour every second day since the automation started. We tried to make our statement clearer by adding the following sentence: "In this calculation all days are taken into account, regardless of whether the measuring conditions were good or bad."

15. L281 - Adding the pollution rose plot for $CO_2$ enhancement of the station inside

the city could also be very interesting here to learn about the source distribution inside the city limits.

**Response:** Thank you for your suggestion. We added the concentration plot for the center station in the figure.

16. L293 – How much less data is available for southern station

    **Response:** We added a new table (Table 1) that includes the dates when the respective instruments started to measure in our network as well as the measured data points taken so far.

17. L296 - This study does NOT show the drastic impact on GHG emissions, but mere a decrease in local GHG enhancements. There are many other possible reasons for changes in GHG concentrations other then emission changes. It is reasonable to assume here that the concentration enhancement change is due to emission changes, but this should be stated carefully and other potential sources of uncertainty have to be included when referring to emissions.

    **Response:** Thanks for your suggestion. We attenuated our statements regarding our ability to determine emissions throughout the whole document. In this section, we changed the headline to "Influences of the COVID-19 lockdown *on urban concentration gradients*". Furthermore, we changed L296 to: "[...] showing the influence of such a drastic event on the *urban GHG gradients* of a city like Munich."

18. L302 - Please provide the R2 for this relationship. Also, looking at figure 14 it seems clear that $CO_2$ enhancements decreased strongly in week 6 and 8 already, well before the lockdown period, while congestion was above 25%, i.e. fairly normal. A scatter plot of the two quantities could be a useful addition in the supplemental information of this paper.

    **Response:** Thanks for this valuable suggestion. We modified our statement regarding the correlation of our measurements to the traffic data and added the R2 value. The new formulation is: "The plot demonstrates that the lockdown had a significant impact on traffic flow. The $CO_2$ enhancements show a similar pattern throughout the first half of the year 2020. Based on the regression plot, there seems to be a correlation between the reduced traffic volume and the lower $CO_2$ enhancements ($R^2$=0.63). Both curves first decrease and then increase again after the strict restrictions were gradually loosened." As per your suggestion, a scatter plot was introduced to Figure 13.

19. L304 - See comment L296, L302, this study does not establish a decrease in emissions within Munich. Further modelling (including biospheric $CO_2$) and assessment of uncertainties seems necessary before the authors should claim that they have proven that their system is sufficient to track emission changes. The authors later refer these uncertainties, so they seem aware of this problem, so why make such a strong claim here? Being able to reliably track $XCO_2$ enhancement changes during COVID lockdown with an automated system is already an excellent achievement in itself.

    **Response:** Thank you. We have changed the statement (see response to comment of L302)

20. L327 - This statement completely ignores the potentially large impact on $CO_2$ concentrations by the urban biosphere, that has been found to be an important $CO_2$ sink (and sometimes source) in urban areas, for example, Miller et al. 2020 (PNAS, https://doi.org/10.1073/pnas.2005253117).

    **Response:** Yes, our statement is too simplistic here. We changed the sentence to "For that, the concentration gradients between the downwind and upwind stations are decisive, as they represent the anthropogenic emissions superimposed with biological processes."

21. L335: No data set of traffic emissions was presented in this paper. I agree that the seen decrease in congestion makes emission reductions extremely likely, but this should be stated carefully. Also the decrease does not seem to be concurrent.

**Response:** We modified our sentence to: "The results show a *possible* correlation between the $CO_2$ column concentration gradients and the *traffic amount*, both of which appear to be drastically affected by the lockdown."

22. L342: It is unclear how this study has established that column measurements can be used in "any city worldwide". It seems apparent that the concentration gradients in the total column for smaller cities might be too small to detect reliably or the $CO_2$ emission signal might be masked due to biospheric uptake in cities in the tropics. What about cities with very strong aerosol loads, like Beijing, would the EM27SUN be able to penetrate dense smog?

**Response:** Thank you for your comment. As mentioned before, we do not claim anymore that we can measure emissions but concentration gradients. We changed it in the paper accordingly. Based on that, we think that our approach can be used in many cities worldwide. We changed "any city" to "over a wide range of latitudes". The statement we want to make is that we developed the sensor system that is necessary to establish a permanent ground-based remote sensing network using EM27/SUN independent on the location.

Furthermore, there exist FTS sites in large Chinese cities such as Beijing (Bi et al., 2018: https://doi.org/10.1007/s13351-018-7118-6) and Hefei (Wang et al., 2017: http://dx.doi.org/10.5194/amt-10-2627-2017).

With best regards,
Florian Dietrich on behalf of all co-authors

[Figure]

**Fig. 1.** Concentration enhancements over the background for each of the five stations displayed as a polar histogram.

[Figure]

**Fig. 2.** Correlations between the CO$_2$ enhancements over the background measured at our inner city station in Munich, and the traffic amount represented by the congestion rate (time series + regression plot)